# Persistent impacts of the 2018 drought on forest disturbance regimes in Europe

Cornelius Senf[1], Rupert Seidl[1,2]

[1]Technical University of Munich, Hans-Carl-von-Carlowitz-Platz 2, 85354 Freising, Germany
[2] Berchtesgaden National Park, Doktorberg 6, 83471, Berchtesgaden, Germany

*Correspondence to*: C. Senf (cornelius.senf@tum.de)

**Abstract.** Europe was affected by an extreme drought in 2018, compounding with an extensive heatwave in the same and subsequent years. Here we provide a first assessment of the impacts this compounding event had on forest disturbance regimes
in Europe. We find that the 2018 drought caused unprecedented levels of forest disturbance across large parts of Europe, persisting up to two years post drought. The 2018 drought pushed forest disturbance regimes in Europe to the edge of their past range of variation, especially in Central and Eastern Europe. Increased levels of forest disturbance were associated with low soil water availability in 2018, and were further modulated by high vapor pressure deficit from 2018 to 2020. We also document the emergence of novel spatiotemporal disturbance patterns following the 2018 drought (i.e., more and larger
disturbances, occurring with higher spatiotemporal autocorrelation) that will have long-lasting impacts on forest structure, and raise concerns about a potential loss of forest resilience. We conclude that the 2018 drought had unprecedented impacts on forest disturbance regimes in Europe, highlighting the urgent need to adapt Europe's forests to a hotter and drier future with more disturbance.

**Main text**

Europe was affected by a severe drought in 2018, characterized by extreme and persistent soil moisture deficits (Peters et al., 2020) and intense heat in 2018 and the following years. The event was consistent with emerging climatic extremes under global change, characterized by prolonged precipitation-free periods coinciding with elevated water loss due to high temperatures during heatwaves (Ault, 2020).
Such combined drought and heat events are thought to be major drivers of forest disturbances through direct tree mortality, and through facilitating insect outbreaks and wildfire (Allen et al., 2015; Brodribb et al., 2020; Seidl et al., 2020). Increased forest disturbances from drought can push ecosystems beyond their historic range of variation (Johnstone et al., 2016), leaving the 'safe operating space' these systems have functioned in for decades to centuries. As a consequence, emerging novel drought regimes pose a
substantial thread to global forest resilience (Trumbore et al., 2015; Millar and Stephenson, 2015).

In Europe, drought is considered a major driver of forest disturbance (Senf et al., 2020), with disturbance here defined as any abrupt decline in the dominant forest canopy. Increased forest disturbance and early leaf-shedding have also been reported in response to the 2018 drought (Schuldt et al., 2020; Brun et al., 2020). However, evidence remains anecdotal and the large-scale effect of the
2018 drought on forest disturbance regimes (i.e., the prevailing spatiotemporal patterns of disturbance) in Europe remains unquantified. We here conducted a first quantitative assessment of the 2018 drought impacts on the forest disturbance regimes in Europe by providing an update of a satellite-based pan-European forest disturbance map (Senf and Seidl, 2021a) until 2020, and by analyzing changes in disturbance regimes following the 2018 drought. We hypothesized that the low soil moisture
availability in 2018 and the high atmospheric water demand in 2018-2020 led to persistent increases in disturbance, which have pushed Europe's forest disturbance regimes to the edge of their past range of variation.

We found a substantial increase (up to +500 % compared to the average of 1986 – 2015; Fig. 1 a) in forest disturbances in large parts of Europe in 2018, which spatially aligned with observed soil
moisture and vapor pressure deficit anomalies in the summer of 2018 (Fig. 1 b/c). The positive disturbance anomaly was persistent beyond 2018, with disturbance rates remaining considerably above average at least until 2020 (Fig. 1). The elevated levels of disturbance observed in 2019 and 2020 were significantly correlated with negative soil moisture anomalies in 2018 (Fig. 2), suggesting that the 2018 drought had persistent impacts on forest disturbances for at least three years. Soil moisture anomalies in
2019 and 2020 were also significantly correlated to disturbance anomalies in those years, but effects were weaker than those of the soil moisture anomalies in 2018 (Table 1). This suggests that drought conditions in 2018 were already indicative of impacts on disturbances observed in the following years. We further found a significant interaction effect between soil moisture anomalies in 2018 and vapor pressure deficit anomalies in 2019 and 2020, but not in 2018 (Fig. 2 and Supplementary Table S1).
Specifically, we found higher positive disturbance anomalies in areas that were affected by both low soil moisture in 2018 and high vapor pressure deficit in 2019 and 2020 (Fig. 2). This result highlights the combined effect of extreme soil moisture deficits and co-occurring atmospheric dryness because of heat, which was characteristic for the drought of 2018 and the following years (Fig. 2 b/c). Overall, summer soil moisture and vapor pressure deficit anomalies alone explained 11.5 % of the total
continental-scale variance in disturbance anomalies for 2018 – 2020. Yet, we note that there is

remaining variability in disturbance not explained by drought and likely related to forest management (Sebald et al., 2021; Senf and Seidl, 2021b), structural drivers (Seidl et al., 2011), and local processes not considered in this analysis (i.e., topography; Senf and Seidl, 2018; Albrich et al., 2020).

**Figure 1: (a)** Forest disturbance anomalies in the years 2018-2020 relative to 1986-2015, estimated from satellite-based disturbance maps across Europe. Anomalies are expressed in percent area change, that is +100% indicates a doubling of the disturbed forest area relative to the average disturbed forest area in the period 1986-2015. Anomalies were calculated at a grid of ~9 km. **(b)** Summer (JJA) soil moisture

anomalies (z-scores) in relation to the period 1986-2015 at the same spatial grain as (a). (c) Summer (JJA) vapor pressure deficit (z-scores) in relation to the period 1986-2015 at the same spatial grain as (a). Background maps are from https://gadm.org.

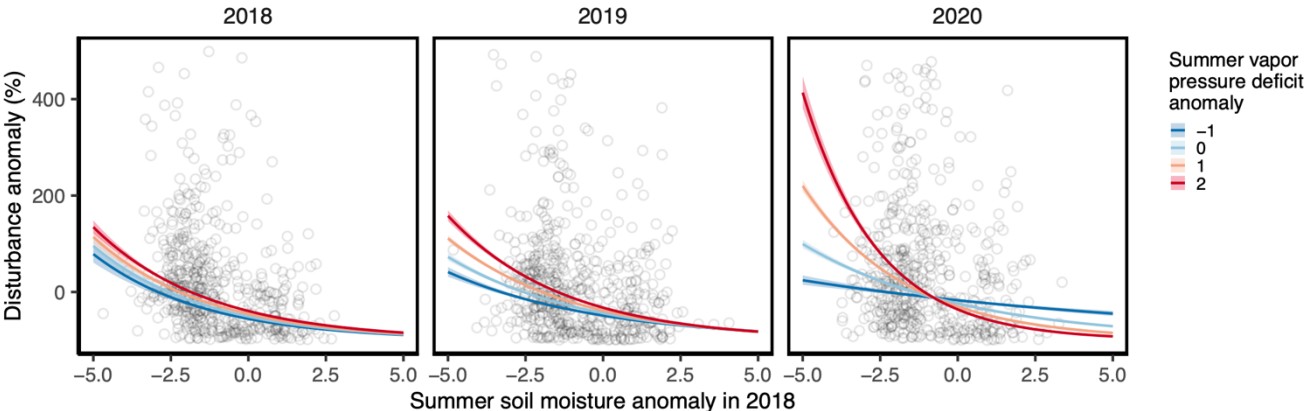

**Figure 2:** Relationship between forest disturbance anomaly in 2018, 2019 and 2020 (see Fig. 1) in relation to local summer (June, July and August) soil moisture anomaly in 2018 and summer vapor pressure deficit (VPD) anomalies in the respective years. All anomalies are expressed relative to the period 1986 – 2015. The black dots show a sample (1 %) of the raw data. Ribbons around solid lines indicate the 95 % confidence interval. Note that disturbance anomalies were capped at +500 to improve visibility. A more detailed version of this figure is available as Supplementary Figure S1.

**Table 1:** Competing models compared for linking soil moisture (*sm*) and vapor pressure deficit (*vpd*) anomalies with disturbance anomalies (*A*) across Europe. The models use soil moisture and vapor pressure deficit from different years (*t*). Models are compared using Akaike's Information Criterion (AIC) with smaller values indicating higher support of the model from the data.

| Competing models | Formulation | AIC |
|---|---|---|
| Soil moisture from 2018 and vapor pressure deficit from 2018 throughout 2020 | $A_{i,t} \sim sm_{i,2018} * vpd_{i,t} * t$ | 542627 |
| Soil moisture and vapor pressure deficit from 2018 | $A_{i,t} \sim sm_{i,2018} * vpd_{i,2018} * t$ | 543067 |
| Soil moisture and vapor pressure deficit from 2018 throughout 2020 | $A_{i,t} \sim sm_{i,t} * vpd_{i,t} * t$ | 548963 |

Based on our assessment, we estimate that approximately 1.56 million hectares of forest were disturbed in Europe in 2018, and that 4.74 million hectares were disturbed over the period 2018 – 2020. This is an average annual surplus of ~360.000 hectares for 2018 – 2020, compared to the average disturbed area in 1986 – 2015. The strongest increase in forest disturbances was observed in Central Europe (Fig 3; mostly Germany, Czechia and Austria; Supplementary Table 2) and Eastern Europe (Fig. 3; Belarus and Ukraine; Supplementary Table 2). Yet also in Northern Europe disturbance rates were among the highest observed over the past 35 years (Fig. 3). In contrast, canopy disturbance rates in Western and Southern Europe, i.e., areas not as strongly affected by the extreme drought of 2018 (Fig. 1 b/c), remained within their recent range of variation (Fig. 3).

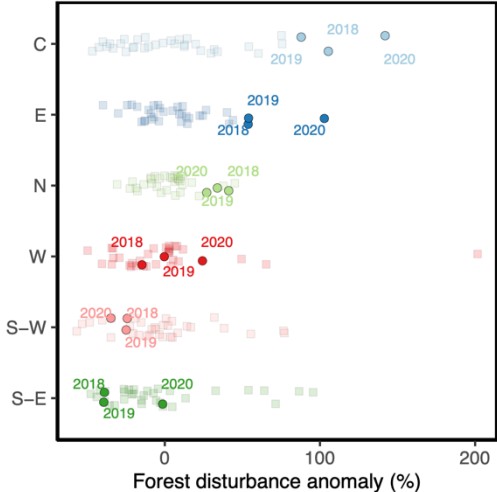

**Figure 3:** Forest disturbance anomalies at the regional level in reference to 1986-2015, with the years 2018-2020 highlighted. Anomalies are expressed in percent area change, that is +100% indicates a doubling of disturbed forest area relative to the average forest area disturbed in the period 1986-2015. Abbreviations for the regions are: C = Central Europe; E = Eastern Europe; N = Northern Europe; S-E = South-Eastern Europe; S-W = South-Western Europe; W = Western Europe. See Supplementary Table 1 for details at the country level.

The persistent and widespread increase in forest disturbances after the 2018 drought suggests that – in addition to direct drought-related tree mortality (Choat et al., 2018) – indirect drought effects in the subsequent years were a major driver of increased disturbances. A particularly important indirect drought effect is the facilitation of insect disturbances (Allen et al., 2015; Seidl et al., 2017). In Central and Eastern Europe, large-scale outbreaks of bark beetles (mostly *Ips typographus* L.) led to a strong increase in infested conifers after 2018. According to national felling statistics, drought and insect activity nearly brought regular forestry to a halt in these regions, with at least 50 % (Austria and Germany) and up to > 90 % (Czechia) of all harvests in 2019 being related to salvage logging (Knížek and Liška, 2020; Destatis, 2020; BMLRT, 2020). Widespread bark beetle mortality also explains the strong increase in forest disturbances in Belarus and Ukraine, where *Ips acuminatus* Gyll. caused widespread pine dieback (Food and Agricultural Organization of the United Nations, 2018). In addition to biotic disturbances, fire activity also increased in the areas affected by the 2018 drought. For example, Finland, Sweden and Norway experienced the highest fire activity on record in 2018, and sharp increases in area burned were also reported for many countries in Central Europe (San-Miguel-Ayanz et al., 2018, 2019). Yet, fire still only plays a minor role in the current forest disturbance regimes of both Central and Northern Europe, and was responsible for only ~ 3 % of the total area disturbed in these areas in 2018. Also, two major storm events occurred in 2018 affecting Poland and Northern Italy, constituting disturbances causally not related to the 2018 drought but emerging in our analysis (Fig 1). These two storms, while being the most extensive pulses of disturbances in the affected regions for many decades, only explained ~80.000 ha of the 1.56 million ha of forest disturbances recorded for 2018 in our analysis.

The persistent increase in forest disturbances reported here will have long-lasting impacts on forest dynamics in Europe. In the past decades, wind was the most important natural disturbance agent

on the continent (Schelhaas et al., 2003; Seidl et al., 2014; Senf and Seidl, 2021b). The single largest forest disturbance event reported in Europe since 1850 was the storm 'Lothar' in the winter of 1999/2000 (Gardiner et al., 2010). We here show that current forest disturbance levels exceeded this past maximum, with levels of forest disturbance being 1.42 times higher in 2020 than in the year 2000

(i.e., the year in which we record the impact of storm 'Lothar'). This indicates that the drought of 2018 might be responsible for one of the biggest pulses of disturbances in Europe in the past 170 years (Schelhaas et al., 2003), though we note that large-scale disturbances also occurred prior to modern records on forest disturbance (Gmelin, 1787).

The recent episode of forest disturbance can have profound and long-lasting impacts on the

structure of Europe's forests. Specifically, we found that not only the amount, but also the size, frequency and aggregation of forest disturbances increased beyond historic levels in 2018 – 2020 (Fig. 4). These attributes are of high relevance for forest dynamics as they shape forest development trajectories for decades to centuries, and are determinants of the resilience of forest ecosystems (Scheffer et al., 2015; Johnstone et al., 2016). While forests have returned swiftly to their historical

attractor after past large-scale perturbations (such as storm 'Lothar' in 1999/2000; Fig. 4), the 2018 drought has pushed forest disturbance regimes in Europe past their basin of attraction for at least three consecutive years, and it remains unclear if the disturbance regime will return within the next years. A continuation of Europe's forests along this new trajectory of increasing frequency, size, and aggregation of disturbances might result in the crossing of tipping-points, causing pervasive and irreversible shifts in

forest ecosystem structure and functioning (McDowell et al., 2020; Anderegg et al., 2012).

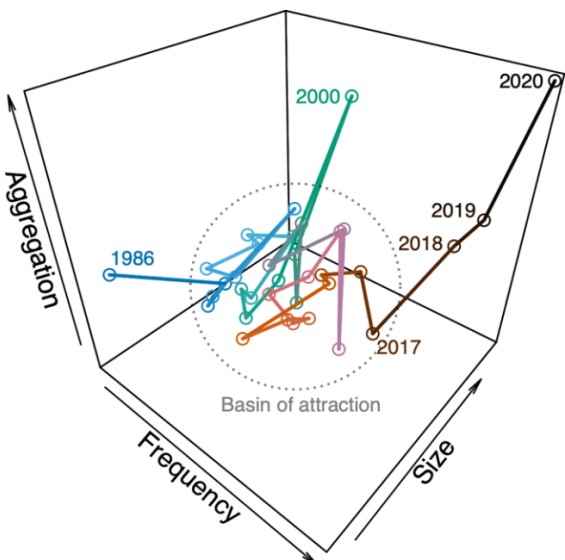

**Figure 4:** The development of disturbance regime characteristics in Europe's forests 1986 – 2020. Frequency denotes the average number
of disturbances per unit forest area and year, size is the 95 % quantile of the patch size distribution of disturbances, and aggregation the
average spatiotemporal autocorrelation of disturbance patches. The drought of 2018 has pushed Europe's forests disturbance regimes
outside of their past basin of attraction.

We here provide a first assessment of the impacts of the 2018 drought on forest disturbance
regimes in Europe. Our analyses of remote sensing data show that forest disturbance regimes in Europe
have changed profoundly following the drought of 2018 and subsequent heatwaves. We note, however,
that satellite-based assessments only provide a coarse-scale view of ecosystem dynamics. Further
research is needed to improve our understanding of the impacts of recent drought and heat events at the
local and regional scale. Our assessment can help to guide these research efforts, and provide
information needed to adapt forests to a hotter and drier future with more disturbance. Future
projections indicate that drought events such as the one observed in 2018 will become the new normal
in the near future (Samaniego et al., 2018; Toreti et al., 2019). Pulses of forest disturbance as observed
in recent years are thus likely also in the coming decades. Hence, we suggest that the causes and
consequences of changing forest disturbance regimes should be a key priority for science and policy.

**Materials and Methods**

We updated an existing pan-European forest disturbance map based on Landsat data, originally
covering the time period 1986-2016 (Senf and Seidl, 2021a), until the year 2020. The map depicts any
abrupt declines in the dominant forest canopy – regardless of its cause – that are detectable at a spatial
grain of 30 m, including disturbances that only remove a part of the canopy within a pixel. It does,
however, not detect any changes in sub-canopy tree layers. In order to update the map until 2020, we
applied the same workflow as used for creating the first version in order to ensure consistency over
time. The initial map product had an overall accuracy of $87.6 \pm 0.5$ % with a disturbance commission
error of $17.1 \pm 1.6$ % and a disturbance omission error of $36.9 \pm 0.02$ %, indicating that the map is
conservative (i.e., higher omission of true disturbances than commission of false disturbances). We
performed a visual quality screening of the map update and did not identify any inconstancies that
might flag a rapid decrease in map accuracy for the recent years. Yet, due to a limited number of clear
satellite observations in Norway for the year 2020, we identified some artefacts stemming from clouds
in the final maps for Norway. To reduce bias in our analysis, we excluded data from Norway in 2020.
The updated map products are available at https://zenodo.org/record/4570157 (Version 1.1.0).
We aggregated the disturbance map from its native 30 m resolution to a regular grid of 0.1° (~ 9
km) by calculating the absolute annual ($t$) area of forest disturbed ($D_{ti}$) per grid cell $i$. From the absolute
annual area disturbed we subsequently calculated the long-term average annual canopy disturbance area
for the period 1986 – 2015 as reference ($D_{i,ref}$), in order to estimate the annual fractional anomaly $A_{it}$
as $A_{it} = D_{it}/D_{i,ref} * 100$. In the following, we refer to $A_{it}$ as annual forest disturbance anomaly per
grid cell. The forest disturbance anomaly is the percent deviation of annual forest area disturbed relative
to the long-term (1986 – 2015) mean. As anomalies can become unreliable when the reference level
$D_{i,ref}$ is very low (i.e., a very small absolute increase can lead to a very large anomaly in such cases),
we excluded all grid cells with < 1 ha of disturbance per year on average from the analysis (excluding n
= 481,770 cells, representing 18 % of all cells). Besides calculating anomalies for each grid cell, we also
calculated them for six European regions, first aggregating annual area disturbed to regional level and
subsequently calculating the anomalies. The regions considered were East (Belarus, Moldova, Ukraine),
Central (Austria, Czechia, Germany, Hungary, Poland, Slovakia, Slovenia, Switzerland), West

(Belgium, France, Ireland, Netherlands, United Kingdom), North (Denmark, Estonia, Finland, Latvia, Lithuania, Norway, Sweden), South-East (Albania, Bosnia and Herzegovina, Bulgaria, Croatia, Greece, Montenegro, Romania, Serbia), and South-West (Italy, Portugal, Spain) Europe.

We also characterized changes in forest disturbance regimes in response to the 2018 drought. Specifically, we calculated the patch size of each individual disturbance patch in Europe (n > 35 million patches), as well as the disturbance frequency (expressed as number of patches per hectare forest area per year). We further characterized the spatiotemporal aggregation of disturbance patches by calculating the proportion of all disturbance patches that occurred in the same year in a five-kilometer radial kernel around each individual disturbance patch. A value of one indicates that all disturbances in close proximity happened in the same year as the focal patch (high spatiotemporal autocorrelation), whereas a value of zero indicates no other disturbances occurred in the same year and in proximity to the focal patch. This measure broadly quantifies the press-pulse-dichotomy of human versus natural disturbance regimes (Sebald et al., 2019). We finally aggregated all three measures to annual values across Europe by calculating the 95th quantile for patch sizes and the average of frequency and spatiotemporal aggregation. We used the 95th percentile for patch sizes instead of the average, as patch-size distributions are highly left-skewed with very heavy right tales, which can obscure the calculation of average patch sizes. The 95th percentile gives a better indication of the width of the patch size distribution than the average.

To assess the impacts of the 2018 drought on disturbances, we used the most recent European Center for Medium-Range Weather Forecast (ECMWF) ERA5-land reanalysis data, which has a spatial resolution of 0.1° (~ 9 km) and is available from 1979 to present (Muñoz-Sabater et al., 2021). ERA5-land has high representativeness of extremes across Europe, especially for soil moisture (Cerlini et al., 2017), which makes it highly suitable for assessing drought impacts on forest disturbances. We extracted the monthly averaged volumetric soil water content from 0 to 289 cm over June to August (Bastos et al., 2020). We scaled the data to anomalies via z-transformation, using the mean and standard deviation of the reference period 1986 – 2015 (following called $sm_{it}$). We further acquired mean temperate and mean dew point temperate for June to August to derive the mean summer vapor pressure deficit following formulas described in Seager et al. (2015) (following called $vpd_{it}$). Using a log-linear model with Gaussian error distribution, we finally modelled the spatial variability in forest disturbance anomalies among grid cells ($A_{it}$) for the years 2018 through 2020 using soil moisture anomalies from 2018 and vapor pressure deficit anomalies from 2018 through 2020. We expected that the soil moisture anomaly of 2018 could explain disturbance anomalies in 2018, 2019 and 2020 due to legacy effects of the 2018 drought on subsequent years. Yet, we also tested models using annual soil moisture (i.e., from 2018 throughout 2020) and vapor pressure deficit from only 2018, and compared them (using Akaike's Information Criterion [AIC]) to the initial model using solely soil moisture from 2018 and vapor pressure deficit from 2018 throughout 2020. We furthermore expected that the strength of association would be significantly modulated by annual vapor pressure deficit anomalies, with simultaneously low soil moisture and high vapor pressure deficit leading to highest disturbance anomalies (i.e., an interaction between soil moisture and vapor pressure deficit). We finally included year as dummy variable to account for differences among years in both the average disturbance anomalies, as well as strength of association between predictors and response. For both the interaction of soil moisture and the inclusion of year as dummy variable we tested whether the model substantially improved in

comparison to a more parsimonious model using AIC. All analyses were performed in the statistical software R (R Core Team, 2020).

**Data and code availability**

All data and code are available under https://github.com/corneliussenf/Drought2018 with a permanent version of this repository as upon acceptance of the article available under
https://doi.org/10.5281/zenodo.5342790.

**Author contribution**

CS and RS designed the research. CS conducted all analysis. CS wrote the manuscript, with comments and revision from RS.

**Competing interest**

The authors declare that they have no conflict of interest.

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
