# Peer review of "Persistent impacts of the 2018 drought on forest disturbance regimes in Europe"

_Biogeosciences, 2021_

## Author Response (AR1)

**Reviewer #1**

This is a highly relevant study, as it addresses spatial and temporal forest disturbances in continental Europe in response to the extreme drought in 2018. The authors draw some important conclusions of the 2018 drought being a lasting trigger of changes in forest disturbances across Europe. While I find this finding very fascinating and relevant, I would wish that the authors could underpin it further. Particularly, the link of low soil moisture/high VPD in 2018 being a main driver of forest disturbances in 2019/2020 should be made clearer. I am convinced that this will increase the relevance of this letter and make it attractive for a large scientific community.

**Response:** We thank the reviewer very much for the positive and thorough review, which helped to improve our study and manuscript.
* * *
Please define forest disturbance (in the context of your study).

**Response:** We agree with the reviewer that a proper definition was missing. We revised the manuscript to include a proper definition of disturbances (L. 31: "[…], with forest disturbances here defined as any abrupt decline in the dominant forest canopy") and disturbances regimes (L. 35: "[…] (i.e., the prevailing spatiotemporal patterns of disturbance) […]").
* * *
Please clearly explain all the components that are considered as disturbance and what degree of forest cover loss is needed for a disturbance to be detected with your RS approach.

**Response:** We agree that more information is needed to fully explain our approach. We added more details on the mapping approach to the methods section (L. 162): "The map depicts any abrupt declines in the dominant forest canopy – regardless of its cause – that are detectable at a spatial grain of 30 m, including disturbances that only remove a part of the canopy within a pixel. It does, however, not detect any changes in sub-canopy tree layers.".
* * *
Was disturbance severity assessed (as in Senf & Seidl 2021)? And if yes, should it not be included in the analysis?

**Response:** We did not assess disturbance severity as in Senf and Seidl 2021 as the focus of our analysis was on the total area disturbed, irrespectively of disturbance severity.
* * *
I think it would be good to understand how an already disturbed area/pixel is treated in the following years post disturbance. Are disturbed areas/pixels considered in the next year of your analysis, e.g. from 2019 to 2020 and do they add to the disturbance rate (e.g. because they are more disturbed the next year) or are they omitted because the forest was already disturbed?

**Response:** We thank the reviewer for this question. Throughout the analysis, a pixel can only be disturbed once, that is a pixel disturbed in 2019 cannot be disturbed again in 2020. This is a limitation of the underlying remote sensing approach, but we argue that this only affects a relatively small proportion of the disturbance areas in Europe. In the initial publication (Senf

and Seidl 2021), we estimated that approximately 10 % of the plots identified as disturbed in the manual interpretation of spectral trajectories were affected by > 1 disturbances (that is less than 2 % of Europe's forested area). From those 'double-disturbances', a large proportion relates to management interventions before a natural disturbance (i.e., thinning, removal of bark beetle-infested trees before salvage logging the whole stand) and might thus appear as a longer-duration disturbance event that ultimately results in a single disturbance patch. We here count those longer-duration disturbances event only once in calculating the disturbance rate to not inflate disturbance rate calculations.
* * *
I think it would be worth to assess potential impacts of low soil moisture and high VPD in 2019 on forest disturbance rates. Eventually it was not the one event in 2018, but a repeated drought/heat that increased disturbance rates... This should be at least check and the results presented.

**Response:** We thank the reviewer for this comment. We agree that drought conditions continued into 2019 and also 2020. To test this, we reran the model using annual soil moisture data – that is soil moisture from 2018, 2019 and 2020, respectively – instead of only soil moisture from 2018. This model showed very similar results as our initial model (see Figure below), though it had slightly lower support by the data (see Table 1 in revised manuscript). We thus consider the initial model as more appropriate for explaining the drought impact on disturbances. That said, we found that for areas of high disturbance anomalies (> 100 %), there was a strong correlation (r = 0.65) between soil moisture in 2018 and 2019. The spatial patterns of soil moisture anomaly in 2018 are thus likely already representative for the persistent drought impacts in 2019. To consider this important point also in our manuscript, we substantially revised the text (see changes below). We also revised Figure 1 to include maps of the soil moisture and vapor pressure deficit anomalies, giving a better idea about the spatial patterns of the drought and how it developed over the years 2019 and 2020. Finally, we included the model comparison in the manuscript (Table 1 in the revised manuscript), present the full results of the model as Supplementary Table 1, and added an additional, more detailed version of Figure 2 as Supplementary Figure S1.

Changes in text (L. 43): "We found a substantial increase (up to +500 % compared to the average of 1986 – 2015; Fig. 1 a) in forest disturbances in large parts of Europe in 2018, which spatially aligned with observed soil moisture and vapor pressure deficit anomalies in the summer of 2018 (Fig. 1 b/c). The positive disturbance anomaly was persistent beyond 2018, with disturbance rates remaining considerably above average at least until 2020 (Fig. 1). The elevated levels of disturbance observed in 2019 and 2020 were significantly correlated with negative soil moisture anomalies in 2018 (Fig. 2), suggesting that the 2018 drought had persistent impacts on forest disturbances for at least three years. Soil moisture anomalies in 2019 and 2020 were also significantly correlated to disturbances anomalies in those years, but effects were weaker than those of the soil moisture anomalies in 2018 (Table 1). This suggests that drought conditions in 2018 were already indicative of impacts on disturbances observed in the following years. We further found a significant interaction effect between soil moisture anomalies in 2018 and vapor pressure deficit anomalies in 2019 and 2020, but not in 2018 (Fig. 2 and Supplementary Table S1). Specifically, we found higher positive disturbance anomalies in areas that were affected by both low soil moisture in 2018 and high vapor pressure deficit in 2019 and 2020 (Fig. 2). This result highlights the combined effect of extreme soil moisture deficits and co-occurring atmospheric dryness because of heat, which was characteristic for the drought of 2018 and the following years (Fig. 2 b/c). Overall, summer soil moisture and vapor pressure deficit anomalies alone explained 11.5 % of the

total continental-scale variance in disturbance anomalies for 2018 – 2020. Yet, we note that there is remaining variability in disturbance not explained by drought and likely related to forest management (Sebald et al., 2021; Senf and Seidl, 2021b), structural drivers (Seidl et al., 2011), and local processes not considered in this analysis (i.e., topography; Senf and Seidl, 2018; Albrich et al., 2020)."

Results from model using annual soil moisture data instead of only soil moisture from 2018:

[Figure]

L32 Why is your approach rapid? Did you left something relevant out?

**Response:** A good comment by the reviewer. 'Rapid' was indeed a wrong description of our approach, as we – of course – did not leave out anything relevant. We thus dropped the word from the manuscript.

L40 Did you check the soil moisture/VPD anomalie in 2019, in many regions this was a dry and hot year too. Particular two subsequent dry years might have been the trigger for disturbances to last. Could you add some additional analysis/ information on this, please?

**Response:** Please see answer above.

L42 To my feeling explaining 11.5% of forest disturbance is not that high, or? Please add some explanation.

**Response:** While we agree with the reviewer that 11.5 % might not sound a lot of explained variance, we note that this is a continental-scale model including only two predictors. Given the high variability in disturbance drivers across all of Europe – 11.5 % of explained variance solely by drought (i.e., soil moisture and vapor pressure deficit) is indeed a substantial proportion. Please also note that this is not only pertaining to the areas affected by drought, but that this is 11.5% of all of Europe's disturbances. We, however, agree that some more context might be needed and we thus added more detail to the text (L. 60): "Yet, we note that there is remaining variability in disturbance not explained by drought and likely related to forest management (Sebald et al., 2021; Senf and Seidl, 2021b), structural drivers (Seidl et al., 2011), and local processes not considered in this analysis (i.e., topography; Senf and Seidl, 2018; Albrich et al., 2020).".

L57 Can you please add some information on how strong the disturbances were (e.g. stand replacing, 50% of forest canopy lost,…)?

**Response:** We thank the reviewer for this question, which addresses and important issue. Based on the maps it is very challenging to estimate the true severity of disturbances, because the severity measure used in Senf and Seidl 2021 is based on the spectral change during disturbance, which only gives an indication of the relative severity. We are currently working on additional analyses converting spectral changes into actual changes in canopy cover (work still under review), but this additional work showed that approximately 75 % of all disturbances in Europe detected in our satellite-based approach were high severity events with > 50 % canopy loss and approximately 10 % of all disturbances in Europe had very high severity (> 90 % canopy loss). This, however, only includes disturbances up to 2016. For the period 2018 to 2020 we estimate the disturbance severity to be higher in many regions, as salvage logging was common in Germany and Czechia, both regions which were historically characterized by relatively low disturbance severities. Hence, as there is no reliable information yet about the proportion of stand-replacing disturbances, we refrained from including those estimates in the manuscript. We however note that our map includes disturbances of variable severity in the revised methods description (L. 162): "The map depicts any abrupt declines in the dominant forest canopy – regardless of its cause – that are detectable at a spatial grain of 30 m, including disturbances that only remove a part of the canopy within a pixel. It does, however, not detect any changes in sub-canopy tree layers.".

L61-62 But you did not directly assess canopy mortality rates, or?

**Response:** No, we refrained from assessing absolute canopy mortality rates based on the maps, as those can be biased due to the higher omission error of the maps. We thus prefer relative statements wherever possible. Future research should use our first assessment as basis for a thorough sample-based assessment of the true, absolute disturbance rates (see, e.g., Senf et al. 2018 [*Nature Communications*] and 2021 [*One Earth*]); but such a sample-based assessment was beyond the scope of this letter, which aims at informing research and management in a timely manner.

Fig 1. Do the presented forest disturbance anomalies per year include the previously disturbed forest patches or were those patches excluded? And how was an increase in disturbance severity addressed?

**Response:** The disturbance anomalies are calculated per year and only include the disturbances occurring in this year. A disturbance occurring in 2018 will thus not inflate the anomaly in 2019 or 2020. Changing disturbance severity was not addressed in this research.

Fig 2 Is this 1% sample an exceptionally good one or was it randomly selected? And do the regression represent all data or the 1% only? I think the summer VPD anomaly is given for 2018. Please add and make this clear in the Figure caption and legend. Please also check the unit for VPD, an anomaly of 5 kPa seems huge, did you mean 0.5 kPa instead?

**Response:** The sample was chosen at random. For a better representation, we added a more detailed version as Supplementary Figure S1. The summer VPD anomaly is from 2018, 2019 and 2020, respectively. The intention was to test for interactions of the 2018 drought (which had, of course, long-lasting impacts into 2019 and 2020; see revised Fig. 1) and heatwaves in 2018 and subsequent years (i.e., 2019 and 2020). This model (i.e., soil moisture from 2018 and VPD from 2018, 2019 and 2020, respectively) also had highest support from the data (see response above and also Table 1 in revised manuscript). We revised the methods section to better describe the details of the model (L. 251ff). Finally, the anomaly in VPD was given in terms of standard deviations (as with soil moisture) and the legend title was incorrect. We revised the figure to correct the legend title.
* * *
L124-125 Add the corrected numbers from your Nature Sust. author corrections.

**Response:** Good catch! We wrote this part of the manuscript before the correction was published. We revised the numbers accordingly.
* * *
L 156 Please add a citation/link for the ECMWF reanalysis data. Could you briefly mention how well ERA5 represents soil moisture?

**Response:** A reference and additional information on the representativeness of soil moisture in ERA5 was added to the manuscript (L. 251): "To assess the impacts of the 2018 drought on disturbances, we used the most recent European Center for Medium-Range Weather Forecast (ECMWF) ERA5-land reanalysis data, which has a spatial resolution of 0.1° (~ 9 km) and is available from 1979 to present (Muñoz-Sabater et al., 2021). ERA5-land has high representativeness of extremes across Europe, especially for soil moisture (Cerlini et al., 2017), which makes it highly suitable for assessing drought impacts on forest disturbances.".

**Reviewer #2**

Senf & Seidl contribute a very interesting assessment of post-drought forest disturbance impacts on European forests. Using a satellite based remote-sensing approach with 30 m x 30 m resolution (I guess with Landsat, not mentioned in the methods), they compute forest disturbance anomalies as the deviation of the 2018-2020 period from a 'long-term' average (1986-2015). They show that anomalies go up to >500%, and conclude that the 2018 drought had "unprecedented impacts on forest disturbance regimes in Europe". The paper is well written and conveys a clear message. Knowing also other works by the authors, I have upmost confidence in the scientific soundness of their analytical approach. My only comment on result reporting is that 'disturbance' is not defined anywhere in the text and it is not clear what a 500% increase actually means. Is this related to changes in forest canopy cover, changes in greenness or something else?

**Response:** We thank the reviewer for the positive and thorough review. We agree that a proper definition of disturbance was missing from the manuscript and we thus revised the main text and methods description to properly define disturbances as used in our study

L. 31: "In Europe, drought is considered a major driver of forest disturbances (Senf et al., 2020), with forest disturbances here defined as any abrupt decline in the dominant forest canopy.".

L. 161: "We updated an existing pan-European forest disturbance map based on Landsat data, originally covering the time period 1986-2016 (Senf and Seidl, 2021a), until the year 2020. The map depicts any abrupt declines in the dominant forest canopy – regardless of its cause – that are detectable at a spatial grain of 30 m, including disturbances that only remove a part of the canopy within a pixel. It does, however, not detect any changes in sub-canopy tree layers".

A positive disturbance anomaly thus indicates a surplus in disturbed area (i.e., a higher disturbed forest area than recorded, on average, in the period 1986-2015). We explain this in detail in the methods section (L. 175ff); yet we revised the figure caption to make this point clearer to the reader (Fig. 1 and 3): "Anomalies are expressed in percent area change, that is +100% indicates a doubling of the disturbed forest area relative to the average disturbed forest area in the period 1986-2015.".
* * *
In my understanding, the conclusion is somewhat overstretching the data, though. Unprecedented impacts can only be defined if the historical level of disturbance is known, which is not the case. The reference to other works (Schelhaas et al. 2003) is not enough to make such a strong statement and the temporal horizon of 170 years mentioned there cannot be taken as a benchmark for precedence. The authors are certainly aware of early reports of large-scale drought-induced forest disturbance in the 18th century ("the wormy drought", Gmelin, Johann Friedrich. 1787. Abhandlung Über Die Wurmtroknis. Leipzig: Verlag der Crusiussischen Buchhandlung) that devastated large forest tracts in the Harz region in central Germany. It is likely that other regions were also strongly affected, but we simply don't know.

**Response:** A valid point by the reviewer which we mostly agree with. We consequently revised the text to tone down our conclusions on the 'unprecedented impacts' of the current disturbance episode. However, we still highlight that the current disturbance episode is the

largest one recorded in the past 35 years, and might be among the largest of the past 170 years, given that the largest pulse of disturbance in that period was reported for the year 2000 (i.e., based on Scheelhaas et al. 2003), and that our values for 2020 clearly exceed our records for 2000. We though agree that for any time before 1850 we simply do not know. The revised paragraph reads as following (L. 119): "The persistent increase in forest disturbances reported here will have long-lasting impacts on forest dynamics in Europe. In the past decades, wind was the most important natural disturbance agent on the continent (Schelhaas et al., 2003; Seidl et al., 2014; Senf and Seidl, 2021b). The single largest forest disturbance event reported in Europe since 1850 was the storm 'Lothar' in the winter of 1999/2000 (Gardiner et al., 2010). We here show that current forest disturbance levels exceeded this past maximum, with levels of forest disturbance being 1.42 times higher in 2020 than in the year 2000 (i.e., the year in which we record the impact of storm 'Lothar'). This indicates that the drought of 2018 might be responsible for one of the biggest pulses of disturbances in Europe in the past 170 years (Schelhaas et al., 2003), though we note that large-scale disturbances also occurred prior to modern records on forest disturbance (Gmelin, 1787).".
* * *
It would have been nice to see how disturbance anomalies relate to other important site factors, like altitude, exposition, soil depth, initial stand density, forest type etc. This would allow insights into climate vulnerabilities of European forests and provide useful information for forest management.

**Response:** While we certainly agree with the reviewer that a more in-depth analysis of disturbance drivers would be of interest, we believe this is beyond the scope of this letter. Moreover, data for many drivers of interest do simply not exist across Europe in sufficient spatial, temporal and thematic detail (e.g., stand density, soil depth, dominant species). We thus refrained from adding additional analysis on further drivers to the manuscript.